# Overhauling CAR T Cells to Improve Efficacy, Safety and Cost

**DOI:** 10.3390/cancers12092360

**Published:** 2020-08-21

**Authors:** Leonardo Chicaybam, Martín H. Bonamino, Adriana Luckow Invitti, Patricia Bortman Rozenchan, Igor de Luna Vieira, Bryan E. Strauss

**Affiliations:** 1Vice Presidência de Pesquisa e Coleções Biológicas, Fundação Instituto Oswaldo Cruz (FIOCRUZ), Rio de Janeiro 21040-900, Brazil; leochicaybam@gmail.com (L.C.); mbonamino@inca.gov.br (M.H.B.); 2Programa de Imunologia e Biologia de Tumores, Coordenação de Pesquisa, Instituto Nacional de Câncer (INCA), Rio de Janeiro 20231-050, Brazil; 3Department of Gynecology, Federal University of São Paulo–Escola Paulista de Medicina (UNIFESP-EPM); Colsan Beneficial Blood Collection Association, São Paulo 04039-032, Brazil; adriana.invitti@gmail.com; 4Celluris, São Paulo 05508-000, Brazil; patricia@celluris.com; 5Centro de Investigação Translacional em Oncologia, Instituto do Câncer do Estado de São Paulo, Faculdade de Medicina, Universidade de São Paulo, São Paulo 01246-000, Brazil; igorluna01@gmail.com

**Keywords:** gene therapy, chimeric antigen receptor, T cell, NK cell, cancer, immunotherapy

## Abstract

Gene therapy is now surpassing 30 years of clinical experience and in that time a variety of approaches has been applied for the treatment of a wide range of pathologies. While the promise of gene therapy was over-stated in the 1990’s, the following decades were met with polar extremes between demonstrable success and devastating setbacks. Currently, the field of gene therapy is enjoying the rewards of overcoming the hurdles that come with turning new ideas into safe and reliable treatments, including for cancer. Among these modalities, the modification of T cells with chimeric antigen receptors (CAR-T cells) has met with clear success and holds great promise for the future treatment of cancer. We detail a series of considerations for the improvement of the CAR-T cell approach, including the design of the CAR, routes of gene transfer, introduction of CARs in natural killer and other cell types, combining the CAR approach with checkpoint blockade or oncolytic viruses, improving pre-clinical models as well as means for reducing cost and, thus, making this technology more widely available. While CAR-T cells serve as a prime example of translating novel ideas into effective treatments, certainly the lessons learned will serve to accelerate the current and future development of gene therapy drugs.

## 1. Introduction

While the first 20 years of gene therapy’s history are filled with advances and setbacks, the era since 2010 has seen a series of clear successes that led to the commercialization of a growing number of products (Table 1), including the first approvals in the European Union as well as in the USA. Interestingly, six of the ten approaches are for the treatment of cancer. Even with these advances, we now face new challenges related to the highly specialized application of several of these approaches, implying that only a few patients are candidates for treatment and that the cost will be high for such focused and personalized approaches. Safety is also a concern as more experience is gained with these new drugs and as they are further developed for additional patient populations.

Undoubtedly, the array of pathologies known collectively as cancer represents one of the costliest burdens on society, whether in terms of human health, quality of life, productivity or financial onus. According to the American Cancer Society, while great strides in prevention and therapeutic approaches have been made in recent years, even resulting in reduction in the number of deaths due to cancer, the search continues for efficacious interventions that may be adequately matched to patient’s particular disease profile [1]. While cancer gene therapy has held a promising position since its inception, recent data indicate truly effective treatments are emerging and approval for commercialization is now delivering on this promise.

The term cancer gene therapy encompasses a wide range of possible strategies that generally fit in two main categories: the direct induction of tumor cell death or the stimulation of an anti-tumor immune response. Some classic examples of using gene therapy to induce tumor cell death include the transfer of the p53 gene as well as suicide gene therapy or anti-angiogenic strategies. Alternatively, gene transfer strategies that provoke a tumor-specific immune response are considered as immunotherapies and include the introduction of immune modulating genes, vaccine strategies, oncolytic viruses and adoptive cell transfer (ACT) following ex vivo gene modification. Here we will highlight the approaches involving chimeric antigen receptors (CAR), including the currently available drugs as well as efforts to expand their application, increase safety and reduce cost.

## 2. Chimeric Antigen Receptor T Cells (Car-T Cells)

While a variety of adoptive cell transfer technologies have been tested [2,3], the recent success with one particular approach, CAR-T cells, has attracted a great amount of attention and deservedly so. In this strategy, the patient’s own T cells are modified to express a recombinant receptor that provides both antigen recognition and stimulatory signals, such that cytolysis is initiated in a direct and definitive manner (Figure 1A–C). Since CAR-based antigen recognition is mediated, typically, by a single chain antibody, there is no need for the antigen to be presented as a peptide by the major histocompatibility complex (MHC). Thus, any cell surface molecule could serve as the target antigen and the lack of HLA (human leukocyte antigen) expression, a common feature of immunoedited tumors, does not hinder this approach. In addition, there is no need for co-stimulatory signals since the CAR carries all necessary receptor domains to promote T cell activation [3].

Remarkable success has been encountered with CD19 specific CAR-T cells, leading the approval by the FDA in late 2017 of two approaches, Kymriah (tisagenlecleucel, CTL019, Novartis, Basel, Switzerland) and Yescarta (axicabtagene ciloleucel, Axi-cel, Gilead/Kite Pharma, Santa Monica, CA, USA) for the treatment of relapsed/refractory B cell acute lymphocytic leukemia (r/r B-ALL) and relapsed/refractory diffuse large B cell lymphoma (r/r DLBCL), respectively [4]. In 2018, the FDA approved the use of Kymriah for the treatment of r/r DLBCL [4]. Since CD19 is expressed on all B cells, application of either Kymriah or Yescarta may lead to the elimination of both normal and transformed B cells, a relatively well tolerated trade off treatable with immunoglobulin replacement. However, the massive expansion and activation of the transgenic T lymphocytes and their interplay with myeloid cells leads to the sudden release of cytokines (known as cytokine release syndrome, CRS) and, as a consequence, inflammation and a series of manifestations, such as fever, cardiac, renal and hepatic dysfunction. Neurotoxicities, termed CAR-T-related encephalopathy syndrome (CRES), may also be present, including delirium, aphasia and seizure. The cause of CRES is not clear but may be due to break down of the blood brain barrier in response to cytokines. While manageable, either of these adverse events may be fatal [4].

The approval of Kymriah for the treatment of r/r B-ALL was based on a Phase II trial where 75 children and young adults were treated. Event free and overall survival at 12 months was 50% and 76%, respectively and median duration of remission was not reached [5]. While Kymriah’s performance for the treatment of lymphoma revealed that 64% of patients had a response, where complete remission was seen in 43% of r/r DLBCL and 71% of follicular lymphoma and sustained remissions were seen in 86% and 89% of these patients, respectively [6]. The clinical testing of Yescarta was performed in 101 patients, yielding objective and complete response rates of 82% and 54%, respectively, and overall rate of survival at 18 months of 52% [7]. These outcomes are quite striking considering that most, if not all, of these patients had failed previous treatments and had turned to CAR-T cell therapy as their last hope.

Even with these advances, many topics require further exploration. As described below, the modification to the CAR may improve safety and permit the use of this approach even when the ideal antigen target may not be available, a situation often encountered in solid tumors. The CAR strategy is not limited to T cells but has been explored in natural killer (NK), macrophage and induced pluripotent stem cells (iPSC), as will be discussed here. The high cost of this approach may be reduced by employing alternative methods for generating the CAR cells, including the route of gene transfer as well as reducing the need for a personalized therapy.

## 3. Improving the Car for Increased Safety and Efficacy

CAR-T therapies are among the most promising treatments for cancer, particularly for refractory lymphoid malignancies. In spite of the proven efficacy of CAR-Ts that recognize CD19 and other antigens, the optimal management for the potentially life-threatening side effects has not yet been achieved [8,9]. The main adverse effects reported in CAR-T clinical trials are cytokine release syndrome (CRS) and neurotoxicity. However, tumor lysis syndrome (TLS), anaphylaxis and on-target/off-tumor toxicity have also been reported [10]. The survival of the CAR-T cells in the patient is also a concern in these therapies [11].

The CAR-mediated T-cell recognition is usually defined by the antibody domain and is independent of MHC presentation. This recognition is extended to any target for which a monoclonal antibody is available [12]. The interactions are strongly influenced by the structure and density of the target molecule on the tumor and the location of the antigen [13]. The CAR’s structure drives the modulation of the transgenic cell’s phenotype, activation status, migratory capacity and tumor recognition, and thus is key to the potency of the approach [14,15].

For these reasons, many strategies have emerged in order to improve the CAR-T therapies, aiming to avoid adverse effects and increase the survival and efficacy of the modified cells. Among them, we cite CARs targeting multiple antigens [12] (Figure 1D,E), controlled/tuned CAR-T, able to modulate patient immune response and/or eliminate the CAR-T cells when a high risk adverse effect occurs [16] (Figure 1F,G); bispecific CARs, that are only activated when two antigens of the tumor are bound [17,18] (Figure 1H); Bi-specific T-cell engagers (BiTEs^TM^, Amgen, Thousand Oaks, CA, USA) that are bispecific antibodies that link the CAR antigen in the T-cell with the tumor antigen [19] (Figure 1I), CAR-T with peptide bridges where a soluble protein links the a “universal” CAR-T cell with the specific tumor target (Figure 1J) or CARs with reduced affinity for the antigen [20].

The application of small molecules to tune CARs, but without cell death, emerged as a good strategy to control CAR function [21]. This approach is very similar to that described for iCaspase9 [22], but does not kill the CAR-T cells. Here the small molecule acts as CAR activator, an “ON-switch”, which can be titrated according to patient response and can be used to control the timing of CAR activation [21]. Even in the presence of the target antigen, the CARs are not switched on if the small activator molecule is absent [23]. The major advantage of this kind of switch is that the CAR-T cells remain in the patient after treatment has been terminated and can be activated when necessary. Unfortunately, efficient mechanisms to revert activated complexes have yet to be described. Other groups have developed reversible systems based on optogenetic activation of T cells for tumor recognition. In these systems, a pulse of blue light enables the antitumor function [24,25].

The mechanism proposed by Wu et al. [21] can be improved using an intercellular small molecule instead of an intracellular one, allowing the construction of a general CAR-T cell, whose specificity will be determined by the small molecule epitope. In this scenario, the CAR-T activity will be strictly dependent upon the formation of a ternary complex (tumor antigen-small molecule-CAR) [26]. To this end, publications describe the use of commercially available monoclonal antibodies [27] and conjugated or chemically modified antibodies [28,29]. This switch model was improved by introducing a peptide bridge in the antibody, thus generating peptide bridge switchable CARs and BiTE™.

“Universal” CAR-T cells, as described in more detail below, refer to the modification of cells from a donor so that they may be applied in a large number of allogenic recipients, thus cutting cost and reducing production time. Here we focus on the technologies that allow the same CAR design to be used in a wide number of situations. The peptide bridge switchable CAR represents a “universal” CAR-T cell that is activated by a tumor-binding Fab molecule that has a genetically engrafted specific peptide. This approach has been proven in vitro and in preclinical models for the treatment of leukemia, breast and pancreatic cancers [26,30,31,32,33]. The switch molecule acts as a bridge crosslinking the CAR-T cells to the tumor cells. The advantages of this kind of switch is that the Fab fragments have a relatively short half-life and the peptide tags have limited immunogenicity [31]. The preclinical results indicate peptide bridge switchable CARs as interesting candidates for clinical trials since the CAR activity can be modulated and switched according to the patient’s response. Also, the CAR cells remain in the patient after treatment termination, supporting the activation of the CAR-T cells in the case of disease relapse. The construction of a “universal” CAR that can be directed to target different disease is also a great advantage [26].

Currently, inducible suicide mechanisms are among the most widely explored means to control CAR-T activation in preclinical studies and avoid the progression of potentially lethal adverse effects, such as CRS and neurotoxicity [11,13,22,34,35,36,37,38]. The main approach, reported in previous works is the inclusion of suicide genes in the CAR cells, such as iCaspase9 [19,34,39], generally activated by a chemical inducer of dimerization (CID). There are some ongoing clinical trials using this strategy [34,37,40,41]. This model of a controlled suicidal inducer has also been successfully applied to mesenchymal stromal cell (MSC) transplantation [42], pluripotent stem cell (iPS) therapies [21] and haploidentical stem cell transplantation [43] in order to eliminate graft-versus-host disease (GVHD). The ability to turn the CAR-T therapy on and off without the need for a new CAR-T cell infusion is desirable as it can reduce treatment costs.

The CAR design has been shown to influence the occurrence of CRS, where the 4-1BB signaling domain is less problematic than CD28 [5,7]. Thus, especially for those constructs utilizing CD28, interventions to control CRS may be required. CRS can generally be managed clinically [35], as it can range in severity from low-grade constitutional symptoms to a high-grade syndrome associated with life-threatening, but rarely fulminant, multi-organ failure [10,14]. Considering this, the use of suicide genes for the ablation of CAR-T cells does not seem to be the best strategy for management of putative adverse effects of the therapy, as it eliminates the therapeutic cells to overcome an event that can be clinically managed. Very few options of in vivo CAR controlling mechanisms which do not kill the therapeutic cells have been described [13] and most of them are specific to the targeted disease, not suitable to other uses.

In this scenario, CARs with the ability to regulate the tumor microenvironment have emerged as more suitable strategies. Blockade of programmed cell death ligand 1 (PD-L1) or its receptor, Programmed Death 1 (PD-1) was described to produce tumor regression in several cancer types, such as lymphoma and lung carcinoma [44,45,46]. The blockage of PD-L1/PD-1 seems to improve the CAR-T therapy, disrupting the immune inhibitory axis and facilitating the action of CAR-T cells [47,48,49]. One way of interfering with the PD-1 signaling circuit in T lymphocytes is to outcompete endogenous PD-1 mediated inhibition by expressing a fusion protein consisting of the extracellular domain of PD-1 and the signaling domains currently used in CARs (i.e., CD28). By doing so, different groups have shown a dominant positive effect on T cell activation, overcoming PD-1 inhibition in T lymphocytes [50,51,52]. T cells expressing PD1:CD28 chimera showed increased effector function, with enhanced production of interferon (IFN)-γ and increased proliferation, suggesting that this approach can effectively convert an inhibitory signal to a positive one.

Along the same lines, but not acting directly on PD-L1, Kagoya et al. [53] have constructed a CAR containing a JAK-STAT signaling domain that exhibits superior in vivo persistence and antitumor effects in both liquid and solid tumors. The domains included in this CAR construct were able to induce and maintain a less differentiated T cell phenotype while showing potent antitumor activity, increasing the effectiveness of CAR treatment [53].

Tocilizumab, an antibody that blocks the interleukin 6-receptor (IL-6R) and thus modulates cytokine expression, is applied to treat CRS in CAR-T therapies and in other diseases related to excessive cytokine expression [54]. Despite the proven efficacy of tocilizumab, a more specific control of IL-6 signaling can lead to better results in clinical outcomes of the CAR therapies, as targeting different sites of the IL-6 signaling complex assembly can yield distinct neutralizing consequences; blocking the IIB site of human IL6R yields a more potent inhibition of IL-6 trans-signaling when compared to the targeting of site I (tocilizumab) under high IL-6 levels [55], increasing the relevance of recent works on IL-6 pathway modulation. Other pathways (e.g., IFN-γ, interleukin 10 (IL-10)) and targets (e.g., JAK family tyrosine kinases and STAT family proteins) can lead to better and more tunable effects of CAR-T therapy acting indirectly on the IL-6 pathway and not only regulating CAR-T action, but also controlling the tumor microenvironment [56,57].

The bispecific CARs also improve CAR efficacy while also modulating the tumor microenvironment, as many combinations of scFv can be made and adapted to the needs of each disease [12]. Grada et al. [58] developed a bispecific CD19 and Her2 tandem CAR that has high efficacy. Also, a CAR targeting carcinoembryonic antigen (CEA) and CD30 have a better performance than monospecific CARs due to enhanced cytotoxic activity of CAR-T cells [59]. Many high efficiency bispecific CARs have been reported in the literature: CAR-PSCA and CAR-MUC1 [60], NGFR-spaced-CD44v6, NGFR-spaced-CD19 and NGFR-spaced-CEA [61]. The main issue with bispecific CARs seems to be the length and choice of the spacer region [6,62,63]. The specific requirements of the spacers or non-antigen binding components of the CAR in the extracellular domain must be carefully chosen and proven in vitro and in vivo [63]. The previous works indicate that even if the spacer domain could provide flexibility for the extracellular domain increasing the distance from membrane, or from other antigen-binding domain, it can impair the T-cell activation, demonstrating that the improvement of binding will not necessarily result in increased CAR signaling [64,65]. Thus, bispecific CARs, by targeting combinations of tumor antigens and/or cellular pathways can improve the CAR efficiency while decreasing its toxicity.

## 4. Alternatives to Primary T Cells

### 4.1. Induced Pluripotent Stem Cells (iPSC)

The development of reprogramming protocols for generation of iPSCs by introducing the transcription factors Oct4, Klf4, c-Myc and Sox2 made possible an unlimited source of cells for regenerative medicine [66], including T lymphocytes. Several reports showed that T cells can be generated from embryonic stem cells and iPSC using OP9 cells expressing Delta-ligand 1 (OP9-DL1) as feeder cells [67,68]. It is also possible to reprogram antigen-specific CD8 T cell clones to iPSCs and differentiating them to T lymphocytes; T cells produced using this protocol showed increased telomere length and maintained TCR (T cell receptor) specificity, indicating this can be a viable approach to generate large numbers of clonal T cells [69,70]. A similar protocol was used for the generation of anti-CD19 CAR-T cells, and these cells showed effective effector function in vitro and in vivo [71]. One important aspect of the protocols described above is that the generated T cells display characteristics of gamma delta T cells or innate lymphoid cells, which express CD3 but also CD56. Recently, a protocol to generate CAR-NK cells from iPSCs was reported, presenting antitumor activity in vivo similar to CAR-T lymphocytes and further increasing the application of this approach [72].

While feasible, there are some features of this technology that need to be resolved prior to its large-scale application. T cells generated using this protocol, if used in the allogeneic setting, can be rejected if the HLA haplotypes are not compatible. To reduce this risk, T cells generated from iPSCs with common HLA haplotypes, obtained in curated biobanks [73], can be used. Also in the allogeneic setting, another risk is the induction of GVHD, which can be minimized by generating iPSCs derived from virus-specific T cells, with the resulting T cells harboring a TCR diversity restricted to viral antigens [74]. Another solution for the GVHD problem is to disrupt the TCR locus and prevent its expression by using genome editing tools [75]. This approach paved the way for the generation of universal, off-the-shelf T cells that will be discussed in below.

### 4.2. Natural Killer (NK) Cells

The successful application of CAR-expressing T cells for cancer treatment prompted the investigation of other cells with cytotoxic activity. Natural killer (NK) cells are cells from the innate compartment with high cytotoxic activity and their activation is finely regulated by a balance between activating and inhibitory signals. These cells can be defined as CD56^+^ CD3^neg^ and can be divided in two main subpopulations: CD56^dim^ CD16^+^ (cytotoxic phenotype) and CD56^bright^ CD16^neg^ (immunoregulatory phenotype) [76]. It is largely accepted that binding of self-HLA class I molecules to KIR (Killer Cell Immunoglobulin-like receptors) inhibitory receptors expressed by NK cells are the main source of inhibitory signals, sparing healthy cells from NK activity, although this cell population has been shown to be inhibited also by immune checkpoint receptors such as PD-1 [77]. As downregulation of HLA class I molecules by tumor cells is a common mechanism of immune evasion, these cells can be targeted by NKs. Moreover, due to the transformation process, tumor cells express stress-induced activating ligands like MICA, MICB (MHC class I polypeptide–related sequence A and B) and ULBPs (UL16 binding proteins), which bind to NKG2D (natural killer group 2 member D) activating receptors and further stimulate NK cells. Some tumor cells shed MIC molecules in their soluble form [78,79,80] or in exosomes [81], saturating NKG2D receptors and dampening T [78,80] and NK cell [81] activation as a mechanism of immune escape.

Moreover, NK cell function can be largely inhibited by immunosuppressive cytokines secreted by the tumor and its accessory cells, altering the expression pattern of activating receptors and NK cell function. The expression of CARs in NK cells can partially overcome some of these barriers by efficiently activating NKs and redirecting the response towards tumor cells. Moreover, due to the presumed limited lifespan in vivo, NK cells have the potential to induce less collateral damage to healthy tissues that share the target antigen (on target, off tumor response). NK cells also have a more restricted cytokine production profile, consisting mainly of IFN-γ and GM-CSF (granulocyte macrophage-colony stimulating factor), and due to its activation mechanism are not expected to induce graft versus host disease (GVHD). All these factors have the potential to increase the safety profile of these cells and constitute potential advantages over adoptive cell therapy using T lymphocytes.

Several studies used the human NK cell line NK-92 as a platform for CAR expression and NK cell therapy advancement. This is a widely used cell line and most groups working with NK cells are familiar with its properties such as the ease of expansion and genetic modification and proven cytotoxic activity against tumor cells in vitro and in vivo [82]. These characteristics, along with the assumed lack of incompatibility with patients, have the potential to turn CAR-NK92 cells into an off the shelf product, lowering the costs of CAR therapy and widening its use [83]. CAR-expressing NK-92 showed improved antitumor responses in preclinical models of chronic lymphocytic leukemia (anti-CD20 CAR) [84], multiple myeloma (anti-CS1 CAR) [85] and neuroblastoma (anti-GD2 CAR) [86], to cite a few. CAR-NK92 cells recognizing CD3 [87] or CD5 [88] were also shown to be an effective strategy against T cell malignancies, where the use of CAR-T cells is hampered by fratricide due to shared expression of target antigen. One study has shown that CAR-expressing NK-92 maintain high effector function even after irradiation, a process that would be necessary in the clinical use of this approach [89]. As for any cell line intended to be used clinically, risks of unrestrained proliferation of the cells in vivo make lethal-dose irradiation of the cell line mandatory to guarantee their inability cause any lymphoproliferative disease. This point is key to ensure the safety of these off the shelf approaches based on cell lines, as is the case for NK-92 derived cell products. Importantly, recent reports showed that CAR-expressing NK-92 cells can be used in combination with drugs like bortezomib [90] or regorafenib [91] or in combination with oncolytic virus [92] to improve responses in solid tumor models. These results led to several clinical trials using CAR-expressing NK-92 cells for patients with CD33^+^ AML (acute myeloid leukemia, ClinicalTrials.gov Identifier: NCT02944162), CD19^+^ (ClinicalTrials.gov Identifier: NCT02892695) or CD7^+^ (ClinicalTrials.gov Identifier: NCT02742727) leukemia/lymphoma and Her2^+^ glioblastoma (ClinicalTrials.gov Identifier: NCT03383978).

Most of the aforementioned CARs were originally designed for T cell activation, bearing zeta chain as signal one and CD28 and/or 4-1BB as co-stimulation. The clear potential of CAR-NK cells for cancer therapy led some groups to develop NK-centric CARs, such as those bearing the DAP10 adaptor [93] or NKG2D-based signaling [94] with improved NK function. Several other NK focused CAR-based therapies are currently being developed and have been extensively reviewed elsewhere [95].

The major obstacle for adoptive NK therapy using primary cells is achieving the required number of cells for therapy. This obstacle was overcome by the development of NK expansion protocols using K562-based antigen presenting cells (APCs) expressing membrane-bound IL-15 [93,96] and/or IL-21 [97], which can be adapted for use in the clinical setting [98,99]. NK cells can also be differentiated from HSCs isolated from umbilical cord blood (UCB) and expanded using a cytokine-based protocol, allowing the generation of NK cells in closed systems and according to GMP (good manufacturing practices) regulations [100]. Alternatively, NK cells can be differentiated from induced pluripotent stem (iPS) cells, showing increased antitumor activity when compared to NKs derived from UCB [101]. Regarding primary NK genetic modification, viral vectors and electroporation are generally used, with studies reporting transduction rates of 43–93% for retrovirus [93] and 85% for mRNA electroporation [98].

Efficient protocols for the genetic modification and expansion of primary NK cells paved the way for preclinical in vivo validation. Several studies showed the potential of primary CAR-expressing NK cells for the treatment of different malignancies, such as CD19^+^ leukemias [77], Her2-positive carcinomas [102], neuroblastoma [103] and lymphoma [103]. Additional genetic modification of CAR-NK cells was recently reported, with cells engineered to secrete IL-15 showing improved antitumor response in vivo using a model of B cell lymphoma [104]. A recent report showed that antitumor response mediated by CAR-NK cells expanded from human PBMCs (peripheral blood mononuclear cells) can be inhibited by upregulation of inhibitory HLA-G receptor, uncovering potential resistance mechanisms associated to this therapy [105]. These studies demonstrate that new approaches and treatment combinations are still being explored and might increase the antitumor response seen with CAR-NK cells in the near future. A clinical trial using NK cells isolated and expanded from cord blood and expressing anti-CD19 CAR is underway for treatment of B cell malignancies (ClinicalTrials.gov Identifier: NCT03056339). A recent report described very good response rates in patients treated with NK cells gene modified with a retroviral vector for the expression of an anti CD19 CAR. The therapy showed no remarkable toxicity while promoting tumor response in most of the 11 patients treated, including non-Hodgkin‘s lymphomas and chronic lymphocytic leukemias [106].

An open question regarding NK-based therapies is whether NK cells would be capable of persisting and developing memory-like responses, a relevant aspect for cancer therapy. The same applies to CAR-NK cells. Recent evidence supports the notion of memory-like recall on secondary NK exposure to viruses [107], supporting a memory profile in NK cells [108]. Furthermore, recently a clinical trial with CAR-NK cells showed the CAR bearing NK cells lingering for up to 12 months after the infusion [106], a much longer persistence than previously thought, suggesting these cells could hold an anti-tumor immune response for long periods.

### 4.3. Other Cell Types as Carriers of CAR

The antitumor response is not mounted only by T lymphocytes, but rather is orchestrated along with other cell types from innate (macrophages, neutrophils) and adaptive immunity (B cells). However, few studies attempted to understand the role of cells other than T lymphocytes or NK cells in adoptive CAR therapy (revised in [109]). By using models where all hematopoietic cells were modified to expresses CARs under the control of a pan-hematopoietic promoter, two studies have characterized the role of non-T cells in vivo. De Oliveira et al. [110] showed that CAR-expressing myeloid cells isolated from mice reconstituted with CAR-transduced HSCs can lyse target cells in vitro. In a second study using a similar approach, Yong et al. [111] showed that the in vivo antitumor response was mainly mediated by T cells, but macrophages also contributed to the effect as their elimination with clodronate liposomes decreased the overall survival of mice. Macrophages can be loaded with CARs aimed to promote phagocytosis of cells displaying the target molecule [112]. In this case, an adenoviral vector was used to establish the CAR-M cells which, in addition to recognizing the tumor antigen, behaved as M1 pro-inflammatory/anti-tumor macrophages resulting in decreased tumor burden in two mouse models of tumor treatment [112]. Further studies are necessary in order to deeply evaluate the role of these subpopulations in the context of CAR therapy and whether a combination therapy with T cells is a feasible approach.

Up to now, we have used the term CAR-T cell as reference to αβ T cells, the most abundant T cell population. However, γδ T cells have also been explored for CAR therapy. The γδ T cells share certain properties with NK and other innate cells and can recognize and eliminate tumor cells. The modification of γδ T cells with CAR is thought to provide not only the CAR-mediated recognition of cancer antigens, but also improved tumor infiltration and rapid cytotoxic response functions of the γδ T cells [113,114]. However, particular attention to the design of the CAR may be required since costimulation in γδ T cells may differ from their αβ counterpart [113].

In opposition to the cytotoxic effect promoted by the transgenic expression of CAR in NK and effector T cells, Tregs can be gene modified with CARs in order to promote tolerance to self-antigens, leading to the control of autoimmune disease [115] or impairing graft rejection [116] in mouse models. The application of such an approach may be valuable to restrain certain inflammation process that increases the risk of cancer development.

## 5. New Routes for Gene Transfer: Bringing the Car to the Cell

When long term expression of the therapeutic gene is required, especially in the cells of the hematopoietic system, retroviral and lentiviral vectors are typically used. In fact, Kymriah relies on lentivirus while Yescarta employs gamma retrovirus [6,7]. Since the viral genome integrates within the host’s chromosomes, allowing the exogenous sequence to be passed to daughter cells, this comes with the risk of activating a cellular oncogene. Clearly, safety related to the gene transfer method is a major concern and continues to be monitored. To this end, the Strauss group [117] has developed a rapid and low cost approach to monitoring the population dynamics of cells transduced with lentivirus. Even so, to the best of our knowledge, no safety problems have arisen related to the use lentivirus or retrovirus in CAR-T cell studies [118]. In a recent report on the potential impact of insertional mutagenesis of the anti CD19 CAR inactivating one copy of TET2 (Tet methylcytosine dioxygenase 2) in a lymphocyte already haplo-insufficient for this gene. The clone with this bi-allelic TET2 dysfunction expanded clonally in the patient, being the major clone responsible for the elimination of the CD19 malignancy [119]. Another study points out that the lentivirus used to establish anti CD22 CAR-T cells was associated with clonal expansion of cells harboring provirus insertion in the CBL (casitas B-lineage lymphoma) proto-oncogene, though the patients’ ALL resolved and the CAR-T cells ceased the expansion. A subsequent CD22-negative relapse suggests a mechanism independent of CAR-T cell function [120]. This indicates, so far, that insertional mutagenesis can generate relevant functional data indicating genome locations where CAR insertion impacts T cell expansion.

The cost of producing clinical grade virus is considerable and contributes to the elevated cost of CAR-T cell production [118]. An interesting and unprecedented issue related to the use of these viruses for the production of CAR-T cells has arisen, specifically that GMP manufacturing cannot keep pace with demand [121]. While these issues may be surmountable, there are other options for introducing the CAR construct in target cells and which may prove to be advantageous.

### 5.1. Transposons for CAR Gene Transfer

Transposons are among the most popular non-viral systems being explored for the introduction of CAR constructs into target cells. Gene transfer using transposons requires two essential components: (i) The transposon comprised of an expression cassette containing the gene of interest flanked by inverted terminal repeats; and (ii) A transposase that will direct the integration of the transposon. Since these components may be encoded by plasmids, their manipulation and clinical production are easier and less costly than that of viral vectors [122,123,124]. In addition, the integration pattern of transposons is more random than seen with lenti or retroviral vectors, thus reducing risks associated with insertional mutagenesis [125]. Even so, methods are being explored for the targeted integration of transposons in ‘safe havens’ within the genome [122]. The plasmids are introduced into the target cells, such as T or NK cells, by electroporation or other methods, such as nucleofection [123]. Since the plasmids remain episomal, they will eventually be lost upon cellular proliferation, thus long term expression is supported by the enzyme-mediated insertion of the transposon [122]. In addition, the use of minicircle DNA, essentially plasmids devoid of sequences unrelated to the transgene expression, including the plasmid backbone, can be used to transfer the transposon and further reduce the risk of unwanted responses [126].

The PiggyBac, Sleeping Beauty and Tol2 transposons are all being explored for the transfer of CAR constructs, with CD19 being the typical target antigen [125,127,128]. Even so, transposon mediated gene transfer of CARs targeting mesothelin [129], CD56 [130], EGFR (epidermal growth factor receptor) [131], CD116 [132], IGF1R (insulin-like growth factor 1 receptor) and ROR1 (receptor tyrosine kinase like orphan receptor 1) [133] and HERV-K (human endogenous retrovirus-K) [134] have been reported. So far, clinical data is available from parallel Phase I trials where Sleeping Beauty was used to transfer a CD19 CAR to T cells which were used for the treatment of a total of 26 patients with advanced non-Hodgkin lymphoma or acute lymphoblastic leukemia. In both trials, the patients first underwent hematopoietic stem cell (HSC) transplantation, but in one protocol 9 patients received autologous HSCs and were then infused with patient derived CAR-T cells, whereas in the second protocol, 19 patients received autologous HSCs and then donor derived CAR-T cells. For the autologous HSC transplantation group, progression free survival at 30 months was 83% (100% overall survival), while the allogeneic group, after 12 months, revealed 53% progression free survival (63% overall survival), all without adverse events or elevation of graft-versus-host disease [128]. In these trials, both the gene transfer technology and the application of CAR-T cells in conjunction with HSC transplantation were shown to be effective and safe.

In recent publications, Chicaybam et al. [135] describe the establishment of CD19 CAR-T cells using the Sleeping Beauty transposon delivered in a plasmid vector using nucleofection. Using this approach, they have shown that the modified primary T cells may be expanded upon exposure to an irradiated lymphoblastoid cell line (L388), thus inducing proliferation of the T cells. The presence of CD19 on L388 provided a selective advantage for the CD19 CAR-T cells. The presence of NK cells during the expansion processes aided in the promotion of T cells bearing the CD62L and CCR7 markers which are important for T cell migration to lymph nodes. This expansion protocol was associated with long term CAR expression and killing of B cell leukemia cells in a mouse model [136]. They have also shown that this approach can be used to reduce the production time from 15 days to 8 days, which may translate into cost savings. Moreover, the accelerated production time was compatible with the generation of functional CD19 CAR-T cells that offered long term persistence and elimination of B cell leukemia in vivo [137]. This approach is also feasible by expanding the T cells with clinical compatible Transact beads. Their group recently reported that CAR-T cells generated with SB CARs against CD19 with this protocol retain central memory phenotype and are highly effective in eliminating human CD19+ ALLs in pre-clinical models [138], Furthermore, the rapid gene transfer obtained by combining SB and electroporation allowed the development of rapid protocols for the generation of CAR-T cells. Their group showed that isolation of PBMCs and genetic modification can be obtained in 4 h, generating CAR-T cells with in vivo activity. Cells generated with this rapid point of care compatible protocol were shown to have the same potency of CAR-T cells generated following in vitro expansion in pre-clinical models [139], indicating that CAR-T cells could be generated with minimal manipulation at the site of treatment using low cost protocols, especially in the allogeneic setting.

### 5.2. Site-Specific Insertion of CAR

As mentioned above, the use of virus or transposon to direct the integration of the CAR construct provides long term expression and comes with a risk of insertional mutagenesis. In addition, the essentially random distribution of exogenous sequences within the genome may result in the juxtaposition of the CAR construct with genomic elements that disfavor stable expression of the CAR. As a result, variegated transgene expression, including transcriptional silencing, would be detrimental to CAR-T cell function. To overcome this problem, genome editing techniques may be harnessed to assure the integration and expression of the CAR sequence in a safe and stable manner.

The Sadelain group [140] has used the CRISPR/Cas9 system to insert a CD19 CAR in the T cell receptor α constant (TRAC) locus. In this way, expression of the CAR sequence would be controlled in a more physiologic manner, essentially the same as the endogenous T cell receptor. The targeting construct was based on an adeno-associated viral vector, but gRNA and Cas9 mRNA were introduced by electroporation. They show that expression was uniform and that the resulting CAR-T cells were actually more potent than those established using retrovirus (RV). The TRAC CAR-T cells showed marked reduced expression of markers of exhaustion (PD-1, LAG3 and TIM3) as compared to RV CAR-T cells. In vitro, they show that the levels of CAR expression under the control of the TRAC locus were favorable as compared to high level, constitutive expression from the RV. Using the TRAC locus, the resulting CAR-T cells show a more rapid recovery after exposure to antigen and that CAR expression was more uniform, whereas the RV CAR-T cells suffered a greater lag and variable expression levels [140]. This study shows both a safe approach to integration as well as the importance of the promoter used to control CAR expression, where physiologic dynamics of receptor expression were favorable.

### 5.3. Introduction of CAR Using mRNA

If integration of the sequence encoding CAR is problematic, then perhaps a non-integrating approach would offer certain benefits, such as the lack of insertional mutagenesis as well as an inherently transient duration of expression. The thinking here is that CAR expression would need to last only long enough for the anti-tumor effect to be carried out and that limited expression may reduce toxicity, such as CRS [141]. In particular, the transfer of mRNA encoding the CAR using electroporation or cationic polymers has been attempted by several groups [142,143].

In one example, CAR-T cells were engineered to recognize a neuroblastoma antigen, GD2, using either transient mRNA transfection or permanent lentiviral transduction [144]. Here, the transient CAR-T cells were effective against a localized tumor in a mouse model, but even with multiple applications, these were not sufficient to impede disseminated tumors. In contrast, the stable CAR-T cells eliminated both localized and disseminated tumor with a single application due, in part, to improved penetration in the tumor mass [144]. A study from the June group [145] showed that multiple applications of transient, anti-mesothelin CAR-T cells could in fact mediate regression in a mouse model of disseminated human mesothelioma. In a phase I clinical trial, anti-mesothelin CAR-T cells were established using mRNA and then infused in six patients for the treatment of metastases of pancreatic carcinoma. Three patients demonstrated stable total metabolic active volume and one showed a 69.2% decrease, including complete reduction in liver metastases, though no improvement in the primary tumor [146]. Electroporated T cells with an anti-CD20 CAR have been tested in spontaneous cases of B cell lymphoma in dogs [147]. While this study was more focused on methodology, they did show that this approach could provide modest, transient anti-tumor activity [147]. The Barrett group [142] has looked at means to improve the duration CAR expression upon mRNA electroporation. Specifically, the in vitro transcribed mRNA included 1-methylpseudouridine, a modified nucleoside known to increase stability, and purification included steps to remove double-stranded RNA which would trigger degradation and anti-viral responses. This approach resulted in increased duration and levels of CAR expression, leading to improved cytotoxic activity [142].

### 5.4. Systemic Administration of CAR Using Nanoparticles

So far in our discussion of the CAR-T cell approach, all of the gene transfer technologies were applied to cells ex vivo. This implies that the patient’s cells are harvested, transported to the production facility where they are modified, expanded and then returned to the patient. This process is cumbersome, costly and not widely available since it relies on adequate GMP-compliant facilities. In striking contrast, the work of Smith et al. [148] shows the feasibility of modifying circulating T cells by systemic administration of nanoparticles. Here, the nanoparticles were comprised of a biodegradable β-amino ester and were targeted to T cells by means of anti-CD3e f(ab’)2 fragments and also contained microtubule-associated sequences and nuclear localization signals, thus assuring the transport of the particle to the nucleus of the transfected T cells. When loaded with plasmid DNA encoding an anti-CD19 CAR within a piggyBac transposon, these particles were injected in the circulation of mice and were shown to specifically target T cells which then gained anti-tumor activity comparable to infused CAR-T cells [148]. A similar approach using lentivirus has also been described [149]. While further refinements may be required, this approach opens up the possibility of systemic gene transfer approaches that, as compared to ex vivo manipulations, may be more cost effective.

## 6. Improved Pre-Clinical Models

Most of the results published using human CAR-T cells were generated and validated using human tumor cell lines grafted in immunodeficient mice. While useful for providing an initial response regarding CAR-T cell function, these models have several disadvantages, like the absence or altered function of immune subpopulations, making it difficult to model the tumor microenvironment, and the interactions between human T cells/tumor cells with the mouse stroma, promoting a non-species specific interaction that can hinder the interpretation of results.

An improvement from the model described above is the use of patient-derived xenografts (PDX). In this model, fragments of a primary tumor are implanted in immunodeficient mice without an in vitro culture stage, thereby preserving most of the tumor heterogeneity. The fragments can be implanted subcutaneously or orthotopically, with the latter being preferred. Studies have demonstrated that a fully-grown PDX can be split and implanted in additional mice across multiple generations without losing composition and phenotype [150].

The recent development of immunodeficient mice expressing the human cytokines M-CSF (macrophage colony stimulating factor), IL-3, GM-CSF and thrombopoietin provided a further improvement in PDX models. By using these mice, human HSCs can be grafted and support the generation of functional human monocytes/macrophages and NK cells [151]. These cells infiltrate the tumors and constitute a better model of tumor microenvironment, allowing the study of CAR-T cell interactions with these subpopulations in vivo. The ideal context for this model would be a full autologous setting; however, it can be difficult to collect CD34^+^, tumor and T cells from the same patient.

Another important aspect of the mouse models is their capacity to predict adverse events related to immunotherapy. The main adverse effect of CAR therapy, CRS, was not anticipated by the available preclinical mouse models. Recently, one group used human CD34^+^ engrafted NSG (nod-scid, IL2 receptor γ chain knockout) mice as host animals for human CAR-T cells in an attempt to model CRS. They showed that myeloid subpopulations, including monocytes and macrophages, are the key populations enrolled in the CRS, identifying IL-1 and IL-6 produced by these cells as critical players in the onset of CRS and neurotoxicity [152,153].

Finally, the use of fully murine, spontaneous tumor models might provide valuable information about the interaction between CAR-expressing cells and the tumor microenvironment (TME), since all the components of the TME can be established allowing the natural history of the disease. This kind of model is the most suited to evaluate the contribution of TME cells to the tumor pathophysiology and eventually the role of immune based treatment combined with chemo or radiotherapy, especially for solid tumors [154].

## 7. Combining Car with Other Immunotherapies

Treating solid tumors with CAR-T cells is especially challenging. Target antigens in solid tumors are rarely exclusive to the transformed tissue, creating the risk of on-target, off-tumor responses. Infiltration and function of the CAR-T cells are also hampered by the immunosuppressive TME. While improvements to the CAR design, as detailed above, may resolve some of the issues related to the interactions with the target antigen, overcoming some of the limitations of the TME may be achieved by combining CARs with additional modalities, such as vaccines, checkpoint blockade and oncolytic viruses.

In order to expand the CAR-T cell population in vivo, a recent study described a vaccine strategy where a single, multi-functional molecule is applied systemically. The molecule, called an amphiphile CAR-T ligand, or amph-ligand, has a lipid domain that directs its interaction with albumin, thus carrying the molecule to the lymph nodes. There, the molecule is bound in the membrane of antigen presenting cells and, by means of a specific moiety such as a peptide or small molecule, activates CAR-T cells in the native lymph node compartment. The authors show that this approach resulted in the expansion of CAR-T cells, liberation of IFN-γ and TNF-α, as well as enhanced killing in different tumor models, including B16 mouse melanoma. Since this vaccine approach does not depend on HLA for antigen presentation, it is compatible with CAR-T approaches [155].

As mentioned previously, inhibition of the PD-1/PD-L1 axis is known to improve CAR-T cell therapy. For example, use of gene editing, such as CRISPR/Cas9, to engineer CAR-T cells that lack PD1 expression have been explored [136,156]. Alternatively, the CARs may be engineered to secrete a protein, such as a single chain antibody, that inhibits PD1 [157]. the combined treatment of CARs together with checkpoint blockade was extensively reviewed recently [158].

Oncolytic viruses trigger the death of cancer cells due to viral replication and/or activation of anti-viral responses. The resulting oncolysis is associated with the liberation of factors that stimulate the adaptive immune response, thus oncolytic viruses stimulate immunogenic cell death and are considered to be part of the cancer immunotherapy arsenal. These vectors may also be armed with functional transgenes that enhance the immune response, oncolysis or virus spread. Recent work has shown that oncolytic viruses and CAR-T cells can be used in combination to provide a combinatorial effect [159,160,161].

In one example, an oncolytic adenovirus was co-administered with a helper-dependent adenovirus that expressed a mini-antibody that blocks PD-L1 [162]. While this approach could reduce tumor volumes in a mouse model of prostate cancer, the addition of a Her2-specific CAR-T cell further reduced tumor progression and held tumors in check for 100 or more days. Strikingly, localized production of the mini-antibody was superior to the administration of anti-PD-L1 IgG in combination with the CAR-T cells [162].

In another example [163], the oncolytic adenovirus was modified to express a bispecific T cell engager (BiTE) targeting EGFR in combination with CAR-T cells directed against the folate receptor α (FR-α). The FR-α CAR-T cells could infiltrate the HCT116 colorectal or Panc1 pancreatic tumors, but did not completely eliminate them. In combination with the oncolytic virus encoding the BiTE, essentially complete tumor elimination was seen due to the engagement of both CAR modified and non-modified T cells with the BiTE. Thus the combined approach could eliminate a heterogeneous tumor cell population by employing an oncolytic adenovirus, BiTE and CAR-T cells [163].

Finally, rAd.sT, an oncolytic adenovirus armed with a soluble transforming growth factor-beta receptor II-Fc fusion protein, which blocks TGF-β (transforming growth factor β) signaling, partially inhibited tumor progression in a mouse model of breast cancer [164]. Similarly, a CAR-T cell directed against mesothelin, Meso-CAR-T, were only partially successful in inhibiting tumor progression. Yet their combination provided a stronger effect and also promoted the expression of of IL-6 and IL-12 in the TME [164].

## 8. Reducing Cost

Currently, the cost of approved anti-CD19 CAR-T cell therapy products are $373,000 (Axicabtagene Ciloleucel; Gilead/Kite Pharma, Foster City, CA, USA) and $475,000 (Tisagenlecleucel; Novartis, Basel, Switzerland) per patient, with a total amount of $1 million per patient when costs with medical staff and hospitalization are included. Such a high price hampers the wide application of the treatment and delays further development of the technology. One way to reduce this price would be to decrease the production cost associated with CAR-T cell therapy. Current methods for CAR-T therapy production involve activation of PBMCs obtained from an apheresis product using anti-CD3/CD28 beads, retroviral or lentiviral transduction and expansion of the transduced cells in bags, a cumbersome process which can take 12–17 days and is dependent on significant amount of work from skilled technicians [37]. As discussed, production of GMP-graded viral vectors is a limiting step in the process and adds a great amount to the final cost. Also, use of retro/lentiviral vectors requires testing for replication-competent retrovirus (RCRs), which further increase the costs of the procedure.

Preclinical and clinical testing of several approaches reveals the potential to decrease the overall cost of CAR-T cell therapy. Recent developments allowed the automation of the entire process of T cell transduction and expansion using the CliniMACS Prodigy System (Miltenyi Biotec, Bergisch Gladbach, Germany) [165], thus reducing labor and decreasing the risk of losing the cell product due to contamination. This system consists of closed tubing and chamber that can be filled and washed automatically once connected to a programmed device. In the chamber, cells are activated with beads, exposed to viral vectors, washed and concentrated in serial steps so they can finally be transferred to downstream processing such as freezing of infusion. Reports show that by using this system the CAR-T cell production can be achieved by different centers while using fewer human resources, showing the feasibility of decentralized production. A similar setup can be used to expand NK cells, facilitating the further use of this approach [166]. Automated systems are currently a tendency in the field for gaining scale and consistency and several other closed systems are under development with different levels of automation. These initiatives are likely to deeply contribute for popularizing the cell and gene therapies in the coming years.

Use of non-viral transposon-based vectors for T cell genetic modification is an already proven technology for generation of CAR-T lymphocytes, with an efficiency compared to retro/lentiviral vectors [167]. Transposon vectors allow integration of the transgene and long-term gene expression, and being plasmid-based, requires only the production of GMP-grade plasmids, reducing the costs of the procedure. A recent report showed that reducing lentivirus-based CAR-T cell production time to 3 or 5 days might increase antitumor response by limiting T cell differentiation and reducing expression of inhibitory receptors [168]. Since, unlike for lenti and retroviruses, no prior activation is required to gene modify the cells using transposons, which are delivered by electroporation, this gene transfer platform allows one step generation of CAR^+^ cells that can be readily applied in the clinical setting skipping risks such as the presence of recombinant viruses. As mentioned above, the Bonamino group [135] is pursuing transposons for the generation of CAR-T cells and we expect that the simplicity and quick manufacturing time will contribute to reduced cost.

One of the reasons for the high cost of approved CAR-T cell products is their personalized approach, where a therapeutic product needs to be generated for each patient. Reports in the literature showed that allogeneic virus-specific T (VST) cells used to treat infections in post-transplant patients do not induce GVHD, and can even be used for alternative recipients in which the allogeneic cells were partially mismatched (“third-party” approach) [169,170]. Thus, allogeneic VST cells can be used as off-the-shelf reagents for the generation of CAR-T cells, simplifying the application of CAR-T therapy. Indeed, the use of allogeneic, donor-derived CAR-T cells was already used in a clinical trial for treatment of B cell malignancies, with antitumor response seen in two of six treated patients [171].

Another highly promising approach that has the potential to decrease costs and allow broad application of CAR-T cell therapy is the generation of off-the-shelf, universal T cells. By use of genome editing tools such as zinc finger nucleases, transcription activator-like effector nucleases (TALEN), or the clustered regularly interspersed short palindromic repeat-Cas9 system (CRISPR/Cas9), genes required for TCR complex expression, like TCR alpha or beta chain, can be deleted, generating a product that will not cause GVHD. A recent work used this strategy to generate CAR-T cells, using CRISPR to introduce CAR transgene in TCR alpha locus, thereby deleting the TCR and placing the CAR under the control of the TCR promoter [140]. A similar approach was already tested in clinical trials, where TCR alpha and CD52 (allowing use of Alemtuzumab for host T cell depletion) loci were disrupted by TALENs and the anti-CD19 CAR was inserted by lentiviral transduction. Three pediatric patients were treated, two of the responded with one of them achieving molecular remission [172].

However, elimination of endogenous TCR only eliminates the risk of GVHD, but the infused cells can still be rejected by the host due to HLA mismatch, so other groups target HLA expression by knocking out the B2-microglobulin molecules [173]. Cells knocked out for class I HLA can turn into good targets for NK rejection, so HLA-G gene transfer has been proposed as a way to protect these HLA knockoutcells from NK mediated elimination [174].

Taken together, many of the novel approaches discussed throughout this review may contribute to reduced costs. Improvements in manufacturing infrastructure, reduced reliance on viral gene transfer, faster manufacturing times, universal or off the shelf treatments, to name just a few examples, continue to be improved and are expected to reduce cost and to broaden the availability of CAR-based therapies.

## 9. Conclusions

Here we have discussed some of the mechanics of the CAR, gene transfer and alternative cell types. To maintain focus, we have not addressed the topic of T cell senescence and anergy [175,176], application in non-cancer pathologies, such as HIV [177], cardiac fibrosis [178], autoimmune disease [179,180], though excellent reviews may be found in the literature.

Clearly, the CAR-T cell approach serves as a positive example of bringing novel modalities to the clinical setting. Even with the success in treating certain B cell lymphomas and leukemias, further development of the technology should bring important advances in the safe and reliable treatment of a variety of cancers. And with time, we expect that costs will be reduced, making this approach more widely available.

## Figures and Tables

**Figure 1 cancers-12-02360-f001:**
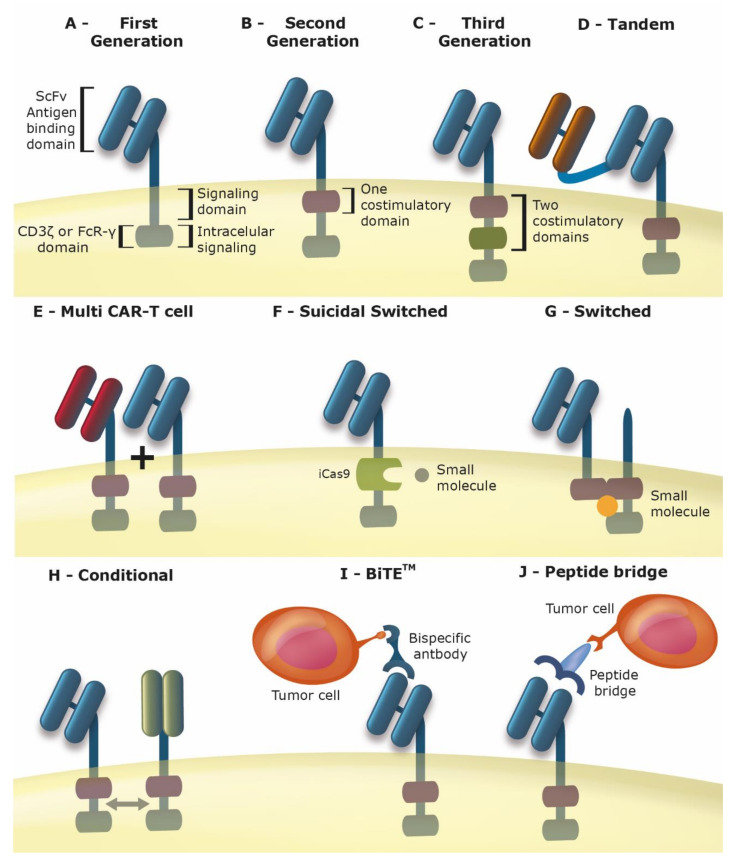
Improving the CAR (chimeric antigen receptors): Schematic representation of alternative CAR designs. (**A**) First Generation CAR-T: extracellular ScFv (single chain fragment variable) domain, transmembrane domain, intracellular signaling domain and CD3ʓ or FCR-γ. (**B**) Second Generation CAR-T: extracellular ScFv domain, transmembrane domain, one intracellular co-stimulatory domain and CD3ʓ or FCR-γ (FC receptor-γ). (**C**) Third Generation CAR-T: extracellular ScFv domain, transmembrane domain, two intracellular co-stimulatory domains and CD3ʓ or FCR-γ. (**D**) Tandem CAR-T–two ScFv domains in tandem connected by a flexible motif or aminoacid sequence. (**E**) Multi CAR-T cell–The same T-cell presents multiple ScFv CARs in the membrane. (**F**) Suicidal Switched CAR-T–Intracellular domain iCasp9 which can be activated by the administration of a small molecule leading to CAR-T cell death. (**G**) Switched CAR-T–The cell has one CAR with a ScFv that transmits the activation signal upon interaction with a small molecule and a second untargeted CAR. (**H**) Conditional CAR-T–Two CAR-Ts that interact between each other and are only activated when both are connected to their antigens. (**I**) BiTE^TM^ (bispecific t cell engager) CAR-T–CAR-T that only ligates to tumoral antigen when a bispecific (CAR-tumor) antibody is administered. (**J**) Peptide bridges CAR-T–Universal CAR-T that will be activated when a peptide bridge links the CAR-T ScFv domain to a cellular antigen, thus the peptide bridge interacts with the ScFV domain as well as the cellular target.

**Table 1 cancers-12-02360-t001:** Principal gene therapy products approved for commercial distribution.

Location (Year/Agency)	Company	Product	Description
China (2003/SFDA)	Shenzhen Gentech SiBiono, Shenzhen, China	Gendicine (Ad-p53)	Non-replicating adenoviral vector expressing wild-type p53 for the treatment of head and neck cancer.
China (2005/SFDA)	Shanghai Sunway Biotech, Pudong, Shanghai, China	Oncorine/H101 (Onyx-015)	Conditionally replicating adenovirus containing a mutant E1b protein which confers tumor-specific oncolysis. Approved for treatment of head and neck cancers.
European Union (2012/EMA)	Amsterdam Molecular Therapeutics, Amsterdam, The Netherlands	Glybera *(alipogene tiparvovec)	Adeno-associated virus encoding the Ser(447)X variant of human lipoprotein lipase (LPL) gene for the treatment of familial LPL deficiency.
USA (2015/FDA) European Union (2015/EMA)	BioVex Inc(a subsidiary ofAmgen, Inc), Woburn, MA, USA	Imlygic (talimogene laherparepvec, T-Vec, oncovex-GMCSF)	Oncolytic herpes virus encoding GM-CSF for the treatment of melanoma.
European Union (2016/EMA)	GlaxoSmithKline (GSK), Uxbridge, Middlesex, UK	Strimvelis (GSK2696273)	Retroviral vector encoding adenosine deaminase (ADA) for the treatment of ADA-SCID.
USA (2017/FDA) European Union (2018/EMA)	Novartis, Basel, Switzerland	Kymriah (tisagenlecleucel, CTL019)	CAR-T cell targeting CD19 for the treatment of B cell acute lymphocytic leukemia.
USA (2017/FDA) European Union (2018/EMA)	Kite Pharma (acquired by Gilead Sciences), Santa Monica, CA, USA	Yescarta (axicabtagene ciloleucel, Axi-cel)	CAR-T cell targeting CD19 for the treatment of diffuse large B cell lymphoma.
USA (2017/FDA) European Union (2018/EMA)	Spark Therapeutics, Philadelphia, PA, USA	Luxturna (Voretigene neparvovec)	AAV encoding RPE65 for the treatment of Leber’s Congenital Amaurosis.
USA (2019/FDA)	Avexis (a subsidiary of Novartis), Bannockburn, IL, USA	Zolgensma (onasemnogene abeparvovec-xioi)	AAV encoding SMN1 for the treatment of spinal muscular atrophy (SMA).

* Withdrawn 2017 due to high costs and low demand. SFDA, State Food and Drug Administration of China; BFAD, Bureau of Food and Drugs; EMA, European Medicines Agency; FDA, Food and Drug Administration, USA. GM-CSF, granulocyte macrophage-colony stimulating factor; SCID, severe combined immunodeficiency; CAR-T cell, chimeric antigen receptor T cell; AAV, adenoassociated virus; RPE65, retinal pigment epithelium-65; SMN1, survival of motor neuron-1.

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
