# Peer review of "Overhauling CAR T Cells to Improve Efficacy, Safety and Cost"

_cancers, 2020, doi:10.3390/cancers12092360_

Round 1

Reviewer 1 Report

Chicaybam et al review the status of CAR T cell technology, alternative methods to improve therapeutical responses to tumors, and ways to decrease costs.  It is a well written review that does a good job summarizing parts of the field. The following points should be addressed in order to improve the impact of the review:

1) Table 1 has a single “(“ after product. Needs to be cleaned up.

2) Table 1 states that Glybera was withdrawn in 2017. Would be useful to include why it was withdrawn.

3) Figure 1 is blurry and difficult to understand in places (e.g With “G” it is not clear from the fitugure why the CAR is only activated when the TCR CAR is bound to the untargeted CAR. Also, “J” could be labeled better to describe the characteristics of the peptide bridge).

4) Figure 1 is only referenced in the manuscript as Figure 1. It would be helpful for the reader if you referenced Figure 1A etc as you describe each construct).

5) Construction of a “universal” CAR is described in line 179, it would be good to give more details at that point on how the universal car is obtained and why it is useful.

6) It would be useful to add a figure illustrating the various strategies to regulate the tumor microenvironment outlined in lines 202-219.

7) How is specific targeting of different sites of the IL-6 signaling complex accomplished? Add these details to the sentence describing this in line 224.

8) In line 229 bi-specific CARs are touted as being able to fill a gap. What is this gap? Need to add context about the gap as the paragraph before is discussing IL-6 specific stratagies.

9) In lines 236-243 papers are cited that have examined the characteristics of linkers for bispecific CAR. It would be helpful to the reader if the general conclusions regarding the linkers from these papers could be included in this paragraph.

10) Line 273 starts a discussion about work using the NK-92 cell line. It would help to include background information on the pros and cons of this cell line for those that do not work with it.

11) If you can include the reason for irradiating the cells in line 283 that would be helpful for the reader. The assumption is they divide continually. If so, are there health concerns if the irradiation is not done correctly.

12) Line 315 discusses NK cell work but does not state if it is in primary cells or a cell line. Please clarify that.

13) Should describe the pros of iPSCs in the paragraph starting in line 320. Also, this whole section seems a bit out of place since the section before and after are discussing cells other than T cells yet section 4.2 is discussing how to reprogram T cells. Consider if there is a better location to move this to.

14) Could not find the reference to De Oliveira et al discussed in line 351.

15) Should update the paragraph on macrophage CARs to contain the newest work Klichinsky, M. et al. Human chimeric antigen receptor macrophages for cancer immunotherapy. Nat Biotechnol. https://doi.org/10.1038/s41587-020-0462-y (2020)

16) Should add a figure for section 5 outlining gene transfer methods and the potential advantages of transposons that are discussed near the end of the review.

17) Would be good to include in the combination section (starting around line 555) the use of vaccines to activate CARs:

Enhanced CAR–T cell activity against solid tumors by vaccine boosting through the chimeric receptor Science  12 Jul 2019:Vol. 365, Issue 6449, pp. 162-168 DOI: 10.1126/science.aav8692

18) In line 603 the CliniMACS Prodegy System automation is described without a reference. It would be good to describe this system more (what it does to save on labor and contamination risks).

Author Response

Manuscript ID: cancers-875295

Chicaybam et al

Response to Reviewer 1

Comments and Suggestions for Authors

1) Table 1 has a single “(“ after product. Needs to be cleaned up.

                This has been addressed.

2) Table 1 states that Glybera was withdrawn in 2017. Would be useful to include why it was withdrawn.

                The withdrawal was due to high costs and low demand. This information was provided in Table I.

3) Figure 1 is blurry and difficult to understand in places (e.g With “G” it is not clear from the fitugure why the CAR is only activated when the TCR CAR is bound to the untargeted CAR. Also, “J” could be labeled better to describe the characteristics of the peptide bridge).

                We have made some modifications to the figure. For panel G, the depiction of the signaling domains was altered. Several spelling errors were corrected. The legend was altered to clarify the role of the peptide bridge in panel J. Finally, we provide a high quality file of the figure (300 dpi, TIFF).

4) Figure 1 is only referenced in the manuscript as Figure 1. It would be helpful for the reader if you referenced Figure 1A etc as you describe each construct).

                The figure panels are now referenced in the text.

5) Construction of a “universal” CAR is described in line 179, it would be good to give more details at that point on how the universal car is obtained and why it is useful.

                Indeed, this point was not clear. Information has been added as suggested.

6) It would be useful to add a figure illustrating the various strategies to regulate the tumor microenvironment outlined in lines 202-219.

                We understand that a figure could be beneficial. However, it would illustrate a single step in a well-known mechanism, so we opted to not include a new figure.

7) How is specific targeting of different sites of the IL-6 signaling complex accomplished? Add these details to the sentence describing this in line 224.

                These details have bee added to the text.

8) In line 229 bi-specific CARs are touted as being able to fill a gap. What is this gap? Need to add context about the gap as the paragraph before is discussing IL-6 specific stratagies.

                This phrase was confusing and has now been changed.

9) In lines 236-243 papers are cited that have examined the characteristics of linkers for bispecific CAR. It would be helpful to the reader if the general conclusions regarding the linkers from these papers could be included in this paragraph.

                A concluding remark has been added.

10) Line 273 starts a discussion about work using the NK-92 cell line. It would help to include background information on the pros and cons of this cell line for those that do not work with it.

Comments regarding NK92’s wide compatibility and the requirement to irradiate the cell line prior to clinical use were included as per request.  

11) If you can include the reason for irradiating the cells in line 283 that would be helpful for the reader. The assumption is they divide continually. If so, are there health concerns if the irradiation is not done correctly.

                As stated above, this information has been included. 

12) Line 315 discusses NK cell work but does not state if it is in primary cells or a cell line. Please clarify that.

                Cells were from primary source and this has been clarified in the text.

13) Should describe the pros of iPSCs in the paragraph starting in line 320. Also, this whole section seems a bit out of place since the section before and after are discussing cells other than T cells yet section 4.2 is discussing how to reprogram T cells. Consider if there is a better location to move this to.

                We have replaced this paragraph as topic 4.1 immediately after describing the primary T cell initiatives. As such, topic 4 was renamed as “4. Alternatives To Primary T Cells”since the iPSCs derived cells are not considered primary T cells. Consequently, the section 4.1 (NK cells) was named 4.2.

14) Could not find the reference to De Oliveira et al discussed in line 351.

                The reference https://doi.org/10.1089/hum.2012.202 was included in the text as requested. Thank you for pointing this out.

15) Should update the paragraph on macrophage CARs to contain the newest work Klichinsky, M. et al. Human chimeric antigen receptor macrophages for cancer immunotherapy. Nat Biotechnol. https://doi.org/10.1038/s41587-020-0462-y (2020)

                We have added some further details pertaining to this work.

16) Should add a figure for section 5 outlining gene transfer methods and the potential advantages of transposons that are discussed near the end of the review.

                We agree that this would be interesting. However, due to time restraints we are unable to build this figure. The citations include additional information on transposon biology.

17) Would be good to include in the combination section (starting around line 555) the use of vaccines to activate CARs:

Enhanced CAR–T cell activity against solid tumors by vaccine boosting through the chimeric receptor Science  12 Jul 2019:Vol. 365, Issue 6449, pp. 162-168 DOI: 10.1126/science.aav8692

                The text now includes a description of this interesting work.

18) In line 603 the CliniMACS Prodegy System automation is described without a reference. It would be good to describe this system more (what it does to save on labor and contamination risks).

                We have included the requested information.

Reviewer 2 Report

This review covers cancer therapy using CAR-modified T cells, describing successes but then some of the pain points as well. The review is well-written, balanced, and uses the references well. The figures are of high quality and appropriate.

There are some small points which I think may further enhance the quality of this already nice work. The authors state that cytokine release syndrome is well manageable, but that is somewhat incorrect: BBz-signaling CARs have a better safety profile than the 28z signaling CARs, and the latter group could certainly benefit from enhanced toxicity management.

This review also covers NK cells equipped with a CAR. What I am missing in that section is the design of NK-centric CARs, i.e. CAR containing signaling domains specifically geared towards the in vivo function of these cells. Also, a paragraph on NK cell memory and how that could be exploited would represent a welcome addition.

In sentence # 375-376 (In fact, Kymriah relies ...Schuster et al., 2017)) I would also use a reference for the Yescarta product rather than two referring to the Kyriah precursor.

Further, aside from the TET2 case Terry Fry and colleagues identified a clonal CAR T cell expansion with CAR insertion into CBL; this case is worth referencing also.

W.r.t. the mRNA-based CAR expression: There are a few recent publications where a mesothelin-specific CAR was introduced as an mRNA; please discuss those also.

Author Response

Manuscript ID: cancers-875295

Chicaybam et al

Response to Reviewer 2

The authors state that cytokine release syndrome is well manageable, but that is somewhat incorrect: BBz-signaling CARs have a better safety profile than the 28z signaling CARs, and the latter group could certainly benefit from enhanced toxicity management.

                This clarification has been included in the text.

This review also covers NK cells equipped with a CAR. What I am missing in that section is the design of NK-centric CARs, i.e. CAR containing signaling domains specifically geared towards the in vivo function of these cells. Also, a paragraph on NK cell memory and how that could be exploited would represent a welcome addition.

                We have now mentioned the CARs designed for NK activation such as those bearing DAP10 and NKG2D domains. We also approached the issue of NK memory and persistence in an additional text inserted in section 4.2.

In sentence # 375-376 (In fact, Kymriah relies ...Schuster et al., 2017)) I would also use a reference for the Yescarta product rather than two referring to the Kyriah precursor.

                The references have been updated.

Further, aside from the TET2 case Terry Fry and colleagues identified a clonal CAR T cell expansion with CAR insertion into CBL; this case is worth referencing also.

                This important information is now included in the text.

W.r.t. the mRNA-based CAR expression: There are a few recent publications where a mesothelin-specific CAR was introduced as an mRNA; please discuss those also.

                We included the phase 1 trial described by Beatty et al.